∂ | Open Peer Review | Bacteriophages | Research Article

# Relevance of the bacteriophage adherence to mucus model for *Pseudomonas aeruginosa* phages

Gabriel Magno de Freitas Almeida,[1] Janne Ravantti,[2] Nino Grdzelishvili,[3,4] Elene Kakabadze,[3] Nata Bakuradze,[3] Elene Javakhishvili,[3] Spyridon Megremis,[5] Nina Chanishvili,[3] Nikolaos Papadopoulos,[5] Lotta-Riina Sundberg[6]

**ABSTRACT** *Pseudomonas aeruginosa* infections are getting increasingly serious as antimicrobial resistance spreads. Phage therapy may be a solution to the problem, especially if improved by current advances on phage-host studies. As a mucosal pathogen, we hypothesize that *P. aeruginosa* and its phages are linked to the bacteriophage adherence to mucus (BAM) model. This means that phage-host interactions could be influenced by mucin presence, impacting the success of phage infections on the *P. aeruginosa* host and consequently leading to the protection of the metazoan host. By using a group of four different phages, we tested three important phenotypes associated with the BAM model: phage binding to mucin, phage growth in mucin-exposed hosts, and the influence of mucin on CRISPR immunity of the bacterium. Three of the tested phages significantly bound to mucin, while two had improved growth rates in mucin-exposed hosts. Improved phage growth was likely the result of phage exploitation of mucin-induced physiological changes in the host. We could not detect CRISPR activity in our system but identified two putative anti-CRISPR proteins coded by the phage. Overall, the differential responses seen for the phages tested show that the same bacterial species can be targeted by mucosal-associated phages or by phages not affected by mucus presence. In conclusion, the BAM model is relevant for phage-bacterium interactions in *P. aeruginosa*, opening new possibilities to improve phage therapy against this important pathogen by considering mucosal interaction dynamics.

**IMPORTANCE** Some bacteriophages are involved in a symbiotic relationship with animals, in which phages held in mucosal surfaces protect them from invading bacteria. *Pseudomonas aeruginosa* is one of the many bacterial pathogens threatening humankind during the current antimicrobial resistance crisis. Here, we have tested whether *P. aeruginosa* and its phages are affected by mucosal conditions. We discovered by using a collection of four phages that, indeed, mucosal interaction dynamics can be seen in this model. Three of the tested phages significantly bound to mucin, while two had improved growth rates in mucin-exposed hosts. These results link *P. aeruginosa* and its phages to the bacteriophage adherence to the mucus model and open opportunities to explore this to improve phage therapy, be it by exploiting the phenotypes detected or by actively selecting mucosal-adapted phages for treatment.

**KEYWORDS** bacteriophage, mucus, Pseudomonas, BAM model

Address correspondence to Gabriel Magno de Freitas Almeida, gabriel.d.almeida@uit.no.

G.M.D.F.A., L.-R.S., and University of Jyväskylä have patented the commercial use of mucin in a patent titled "Improved methods and culture media for production, quantification, and isolation of bacteriophages" (FI20185086, PCT/FI2019/050073).

See the funding table on p. 11.

*Pseudomonas aeruginosa* is an opportunistic bacterium capable of causing disease in vulnerable patients (1). It is one of the pathogens that pose a risk factor for recurrent and chronic airway infections in patients suffering from cystic fibrosis, an autosomal recessive genetic disorder that results in thicker mucus production (1). *P. aeruginosa* prevalence and remarkable antibiotic resistance contributed to make it become one of the ESKAPE pathogens defined by the World Health Organization (1).

One strategy to deal with the increasing antimicrobial resistance threat is the use of bacteriophage (phage) therapy. Phages capable of infecting *P. aeruginosa* are known since the middle of the 20th century, and there have been many studies showing their efficacy as agents of phage therapy in different models (2). *P. aeruginosa* phages have also been used clinically to treat patients, either in compassionate cases (3–5) or by phage therapy centers like the Eliava Phage Therapy Center in Tbilisi, Georgia (6). *P. aeruginosa* was also the target of the Phagoburn clinical trial (7).

*P. aeruginosa* pathogenesis and association with cystic fibrosis make it an example of a mucosal pathogen, and indeed, mucus has been associated with different phenotypic responses for this bacterium. *P. aeruginosa* lipopolysaccharides have been shown to upregulate *muc2* transcription, linking the bacterial presence to overproduction of mucus in patients (8), which reduces the effectiveness of antibiotics (9). *P. aeruginosa* cannot utilize mucin as sole carbon source in single species cultures but likely benefit from propionate generated by mucin-fermenting bacteria in natural settings (10). However, more importantly, mucins may be related to the establishment of invasive phenotypes in *P. aeruginosa*. It has been shown that even though *P. aeruginosa* growth is reduced in mucin solutions compared to growth media, pre-exposure to mucin increases pathogenesis in a keratitis model in rabbits (11). Mucin has also been associated with a specialized swarming motility form (12). Mucin exposure disperses cells from biofilms in a flagella-dependent way, one more hint that mucins are regulators of virulence in *P. aeruginosa* (13). However, when mucin-coated surfaces were tested, they were shown to block motility, leading to large biofilms and increased tolerance to tobramycin (14). The induction of biofilm formation by *P. aeruginosa* following mucin exposure was also seen in a tissue-engineered human lung model, with biofilm forming in a matter of hours (15). Monosaccharides derived from mucin may also affect positively *P. aeruginosa* adhesion and colonization (16). However, in an artificial model of native mucus, *P. aeruginosa* virulence was shown to be downregulated by mucin glycans (17), highlighting that the mucin influence on *P. aeruginosa* physiology can be complex and context-dependant.

Recent evidence has pointed out that certain phages and metazoans may have a symbiotic relationship, in which phages held in mucosal layers have increased chances of finding their bacterial hosts and by doing so they protect the metazoans from invading bacterial pathogens (18, 19). This bacteriophage adherence to mucus (BAM) model has been explored using the mucosal pathogen *Flavobacterium columnare* and shown to be relevant for the prevention of bacterial diseases (20). In this system, it was discovered that phage adhesion to mucus does not only mediate protection from bacterial infections, but mucin exposure also makes the bacterium more prone to phage infections. This suggests that phage infection efficiency is increased in the mucosal environment. A follow-up study added another layer of complexity on phage-bacteria interactions in the mucosal environment linking bacterial countermeasure to the BAM model: mucin exposure enhanced CRISPR spacer acquisition (21). It is possible that selection toward CRISPR defence allows the bacterium to remain virulent during mucosa invasion while being protected from phages therein. We have also shown that mucosal conditions affect phage susceptibility in *Streptococcus mutans* (22).

It is likely that the impact of mucosal interactions is widespread and relevant for other mucosal pathogens besides the ones so far directly linked to the BAM model. In this sense, *P. aeruginosa* and some of its phages may also be involved in the BAM model, but so far, it has not been shown. Here, we used a collection of four *P. aeruginosa* phages isolated at the Eliava Institute and explored how they interact with their host in mucosal conditions, focusing on phenotypes associated with mucin exposure: the ability of phages to bind to mucin and improved phage growth in mucin-exposed hosts. We could see that most of the tested phages are able to bind to mucin and detected a differential response regarding growth in mucin-exposed hosts. We also attempted to evaluate the upregulation of CRISPR activity in *P. aeruginosa* exposed to mucin, but the results were negative. Taken together, our data tie *P. aeruginosa* phages to the BAM

model and offer potential to explore mucosal interactions to better understand mucosal pathogens and improve clinical phage therapy.

## RESULTS

### Details about the phages and hosts used in this study

For this study, we chose four phages isolated at the Eliava Institute in Georgia. The criteria for selecting these phages were their broad activity against *P. aeruginosa* clinical strains and their ability to form a halo around the lytic zone, indicating their depolymerase activity. Details about the four phages used and their characterization (genomes, morphology, host range, and bioinformatics analyses including the search for Ig-like domains) are shown in the supplementary text. The screening of CRISPR-Cas spacers and prophages present in the *P. aeruginosa* strain 573 used for phage isolation is also shown in the supplementary text.

### *P. aeruginosa* phages can bind to mucin

Phage adhesion to mucin has been shown to be mediated by Ig-like domains present in capsid proteins (18, 19). Bioinformatic analyses revealed Ig-binding domains and carbohydrate-binding domains in all four phages used in this study (supplementary text). The mucin-binding phenotype is a strong indication that a phage has relevance for the BAM model. For three of the tested phages, we were able to measure a significant adhesion to mucin (GEC_MRC, $P = 0.0003$; GEC_K2, $P = 0.0302$; and GEC_PNG3, $P = 0.0233$) when comparing phage retention in mucin containing plates to control plates. Phage GEC_PNG14 adhesion to mucin was not significantly different from control plates ($P = 0.0576$) (Fig. 1A). When a ratio was calculated dividing the average of plaque counts in mucin plates by the average of plaque counts in control plates, all four phages shown a trend to have more plaques in mucin plates (Fig. 1B).

### Exposing *P. aeruginosa* to mucin improves replication of phages GEC_PNG3 and GEC_PNG14

Enhanced phage growth in a host pre-exposed to mucin is a second phenotype that can be associated with relevance in the BAM model (20). All the four phages tested grew in their *P. aeruginosa* 573 host, as expected (Fig. 2A through D). However, two of these phages grew more efficiently when the host was exposed to mucin for three hours before infection. Phage GEC_PNG3 titer grows then decreases in control conditions but increases and remains high at the 44 h time point if the host was exposed to mucin ($P =$

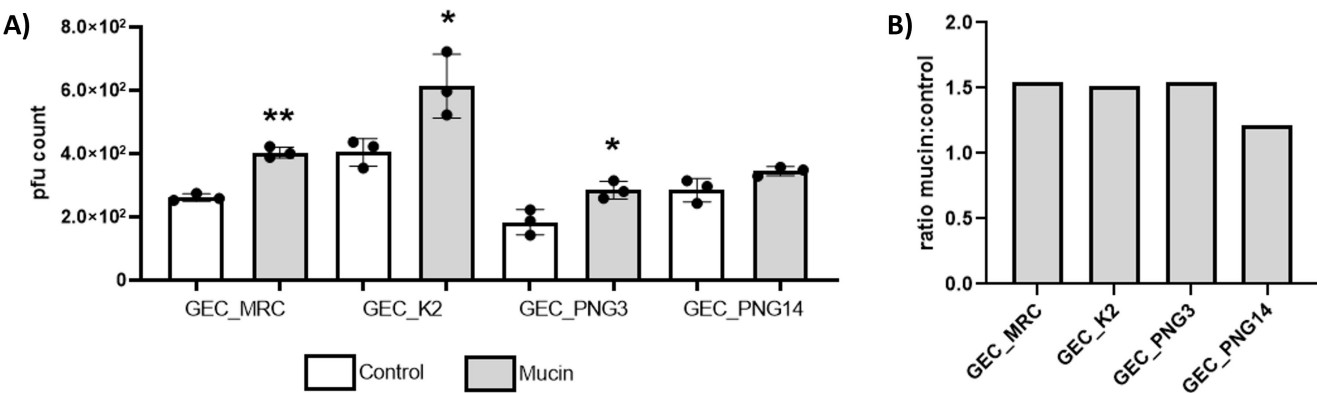

**FIG 1** Phage adhesion to mucin. (A) Plaque-forming unit (pfu) counts from control plates (white bars) and mucin plates (gray bars) are shown for all phages. Each condition was tested in triplicates. Each data point represents one replicate, and the standard deviation is shown in each bar. Unpaired *t* tests were used to compare controls and tested conditions (*$P < 0.05$; **$P < 0.001$). (B) Ratio of plaque counts in mucin plates divided by the average of plaque counts in control plates for all tested phages.

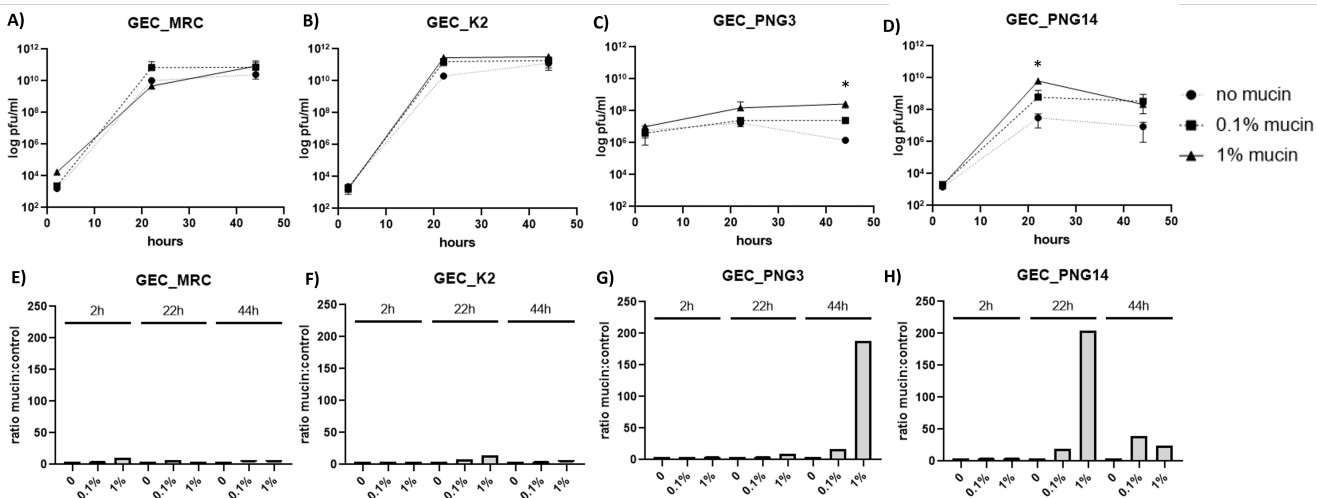

**FIG 2** Phage growth in the presence of increasing concentrations of mucin (0%, 0.1%, or 1%). In A–D, the growth curves of each phage are shown. Each data point represents the average of triplicates and their standard deviation. Unpaired *t* tests were used to compare controls and tested conditions (*$P < 0.05$). E–H are showing the ratios obtained by dividing the average pfu/mL count in mucin conditions by the average pfu/mL count in controls (no mucin), using the averages from the data points in A–D.

0.0164) (Fig. 2C and G). Phage GEC_PNG14 also grew best in mucin-exposed hosts (Fig. 2D and H), but with a growth peak at the 22 h time point ($P = 0.0014$).

## Mucin effect for bacterial physiology is responsible for improved phage growth

Two hypotheses could explain why some phages replicate more efficiently in hosts exposed to mucin. The first is that mucin exposure changes the bacterial host by making it more susceptible for phage infections, as already described for other phage-host pairs (20). The second possibility is that mucin, by being used as nutrient, allows for increased bacterial growth rate and enhanced phage replication is caused by higher availability of host cells to infect. Although the latter is less likely in front of the data showing that not all phages benefit from mucin-exposed hosts, we tested whether mucin had a significant effect for the *P. aeruginosa* 573 host growth. Cultures were prepared exactly as described for the phage growth curves, but no phages were added. Samplings were made at the same time points, but the cultures were serially diluted straight away and plated without the addition of chloroform, for the counting of colony-forming units (cfu). The effect of mucin for the host growth was small, as shown in Fig. 3A and E.

We also tested whether the multiplicity of infection (moi) would affect phage growth. We chose phage GEC_PNG3 as model and the 44 h time point as the readout point considering the results from Fig. 2. Cultures were prepared as described previously and infected with $10^6$ (high moi) or $10^2$ (low moi) plaque-forming units. Phage GEC_PNG3 growth was more prominent in mucin cultures for both moi tested (Fig. 3B). Although a high ratio was seen in the high moi test (Fig. 3F), this was deemed not significative ($P = 0.1564$) likely because an outlier in the mucin treated condition. For the low moi, the difference was significant ($P = 0.0017$, Fig. 3B).

Next, we explored whether using another *P. aeruginosa* host would still result in enhanced phage replication. This test was done to see if the phenomenon is linked to *P. aeruginosa* 573 specifically, or valid for other strains as well. For this, we used the phage GEC_PNG14 as it is the only one of the four phages capable of infecting the *P. aeruginosa* PA14 strain. A similar phage replication experiment was made as the ones presented in Fig. 2 but using the *P. aeruginosa* PA14 strain as host, culture media supplemented with 0% or 0.1% mucin, and the phage and the bacterium were enumerated 18 h after infection. Also in this bacterial strain, exposure of the host to mucin caused a significant

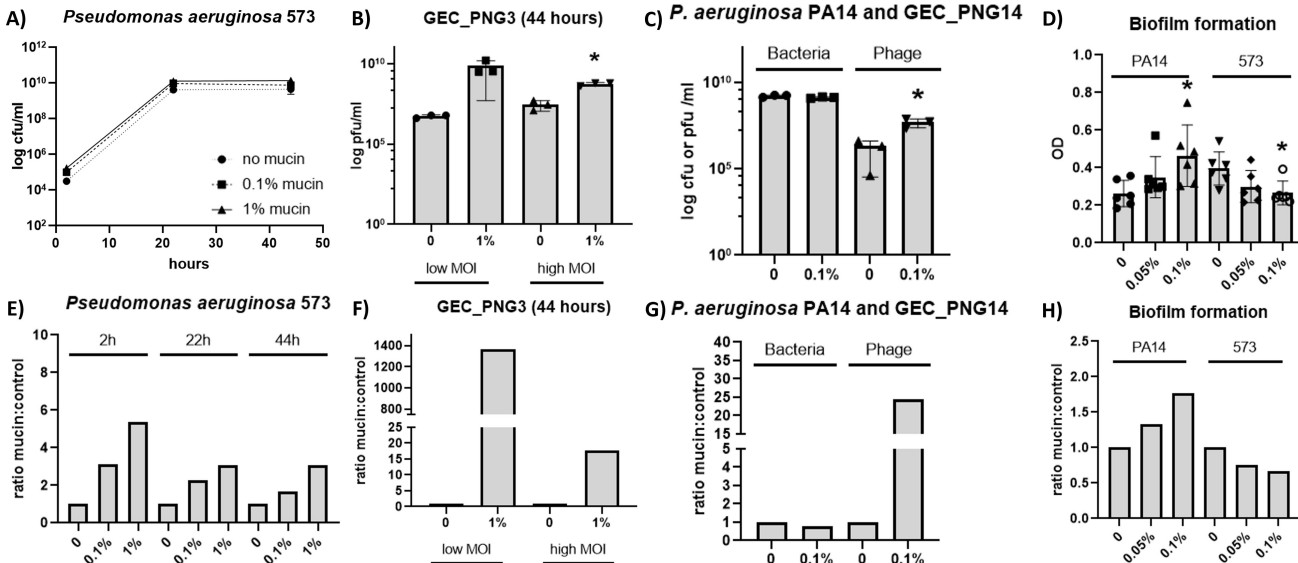

**FIG 3** Testing the effect of mucin in *P. aeruginosa*. (A) *P. aeruginosa* 573 growth curves in increasingly mucin concentrations. (B) Phage GEC_PNG3 growth in *P. aeruginosa* 573 infected with a low or high moi in the presence or absence of mucin (44 h time point). (C) *P. aeruginosa* PA14 host and phage GEC_PNG14 growth in the presence or absence of mucin (18 h time point). (D) Comparison of biofilm formation in increasing concentrations of mucin for *P. aeruginosa* 573 and *P. aeruginosa* PA14. (E-H) Ratios of mucin:control conditions calculated for A, B, C, D, respectively, using the averages from the data points. Unpaired *t* tests were used to compare controls and tested conditions (*$P < 0.05$).

increase in GEC_PNG14 titer ($P = 0.0352$) without affecting the host replication ($P = 0.0964$) (Fig. 3C and G). These results support the suggestion that the improved phage replication is, indeed, a result of cell physiology and not caused by changes in the host population size.

These results exemplify that mucin influences phage replication in two *P. aeruginosa* hosts (PA14 and 573), suggesting that the phenomenon may be widespread and not strain-dependent. Since the exact pathways of mucin sensing in bacteria are not yet known, we evaluated an indirect measurement of mucin effect on bacterial physiology: biofilm formation. The appearance of biofilm in liquid cultures after mucin exposure has been used by us as a "tell-tale" signal that mucin affects bacterial phenotypes. When *P. aeruginosa* PA14 and *P. aeruginosa* 573 were compared, a contradicting pattern appeared. When compared to controls, exposure to 0.1% mucin positively modulated biofilm formation on PA14 ($P = 0.0198$) while influencing negatively the 573 host ($P = 0.0147$) (Fig. 3D and H). This shows that, indeed, responses to mucin can be varied within the same species and that biofilm formation after mucin exposure is not directly linked to improved phage growth.

## CRISPR spacer acquisition

A third phenotype associated with the BAM model is increased CRISPR activity in bacteria exposed to mucin, likely an escape mechanism employed to evade mucosal-associated phages (21). We tested this using phage GEC_PNG14 and the *P. aeruginosa* PA14 host since this host is used as a model for CRISPR studies, and this is the only phage we had capable of infecting it. Long-term (1 month) co-culturing with weekly samplings in four independent experiments was done, all comparing control conditions to mucin presence. Each experiment focused on different parameters that affects CRISPR activity: the first was focused on long-term co-culturing in normal conditions (37°C), the second evaluated low nutrient conditions, the third tested a lower (25°C) temperature, and the fourth combined 25°C to co-culturing with *Flavobacterium columnare*. We were not able to detect any spacer acquisition in any of the sampling on all four experiments (data not shown).

## DISCUSSION

Bacterial infections are becoming increasingly harmful and more difficult to treat as there has been a surge in infections caused by antimicrobial-resistant strains (23). It has been anticipated that by 2050 antibiotic-resistant infections will cause annually over 10 million deaths globally, exceeding deaths related to cancer (8.2 million per year) (24). Worryingly, the latest research estimates that antibiotic-resistant infections caused directly 1.27 million deaths in 2019 and were associated with 4.95 million deaths. Of these, 3.57 million deaths were attributed to infections by *Escherichia coli, Staphylococcus aureus, Klebsiella pneumoniae, Streptococcus pneumoniae, Acinetobacter baumannii*, and *P. aeruginosa* (23). Solutions to this crisis are urgent, and one promising alternative is the use of phage therapy. Although the clinical use of phages is not a new concept, it still can be improved by taking into consideration information derived from modern phage-bacterium interaction studies. One of the recent advancements in this field has been the description of the BAM model (18) and how it can be applied to preventive phage therapy against mucosal infections (20). *P. aeruginosa* is one of the most important causative agents of (antibiotic resistant) hospital-acquired pneumonia, and thus, a clinically pertinent target for phage therapy. Due to its interaction with lung mucosal surfaces, *P. aeruginosa*, classified as one of the ESKAPE pathogens, is likely relevant also for the BAM model. Thus, a detailed understanding of the mucosal interactions between *P. aeruginosa* and its phages can be valuable.

Studies considering how the mucosal environment affects phage-bacteria interaction dynamics are relatively new, likely starting when the BAM model was first proposed and phage adhesion to mucin was shown experimentally using phage T4 (18). To the best of our knowledge, the finding that phages grow best when hosts are exposed to mucin has been described by us recently (20). There is no clear mechanism yet to describe this phenotype even though it is likely that phages adapted to mucosal surfaces have evolved ways to better infect bacteria during the mucosal invasion process. Some hints of this can also be seen in the literature. For example, references (25, 25) describe a mucin-enhanced phage-infecting *E. coli*, while reference (26, 26) reveal that a phage-infecting *Clostridioides difficile* is more efficient in killing its host in the presence of eukaryotic cells (HT-29, known to produce mucins). In a recent study, the transcriptome of *P. aeruginosa* adhered to NuLi-1 epithelial cells and infected with phage LUZ19 was compared to other growth conditions. In this study, host transcriptome varied among conditions, while phage transcriptome remained conserved (27). Interestingly, NuLi-1 cells are *Muc1* producers, and this has already been shown to impact *P. aeruginosa* adhesion (28), indicating that mucin might be the responsible factor for differential gene expression.

Here, we evaluated how *P. aeruginosa* and four of its phages respond to mucin. We focused on the ability of phages to bind to mucins and phage replication in mucin-sup-plemented cultures as phenotypes associated with the BAM model. We observed that mucin influences both the phages and the host. Three phages showed the ignificantly elevated ability to bind on mucin-coated plates (compared to control plates), although bioinformatic analyses indicated that all four phages had hits to Ig-binding domains suggested to mediate phage-mucin interactions, besides also possessing predicted hits to carbohydrate binding domains. Furthermore, two of the four phages benefitted from host exposure to mucins (increased phage replication). This is likely related to changes bacterial cell physiology in mucosal environment. We also attempted to evaluate the mucin effect on *P. aeruginosa* CRISPR spacer acquisition rate, a phenotype associated with mucin exposure shown by us in another mucosal pathogen model (21). However, we could not detect any spacer acquisition in *P. aeruginosa* regardless of the strategy used. It is possible that CRISPR immunity was not observed due to the short-term experimental culture conditions used or due to the activity of the putative anti-CRISPR genes observed in phage PNG14.

The differential responses for the phages tested point to two interesting findings. The first is that phage GEC_PNG14 is likely not a mucin binder but benefits from

host exposure to mucin for growing. This shows that phenotypes linked to mucosal conditions are not always linked. The second is that phages GEC_MRC and GEC_PNG3, despite having 95% nucleotide identity (>90% ORF identity), differ regarding growth on mucin-exposed hosts. These two phages were isolated from different sources, have slightly different plaque sizes, and vary slightly in host range. Exploring their peculiarities might become a good way to understand the mechanisms behind improved growth in mucin-exposed hosts. However, since only four phages were used in this study, we cannot fully determine to which extent mucosal interactions affect *P. aeruginosa* phages. This is a question that needs to be addressed using a larger phage collection. There is also the need to consider the effect of mucin on the bacterial host since we were also able to determine that for one strain mucin had a positive effect in biofilm formation while for the other mucin inhibited it even though these differences did not change the improved growth of phage GEC_PNG14 in the strains. This indicates that, at least for the phages tested here, improved phage growth is not linked to biofilm dynamics caused by mucin exposure of the host bacterium.

Our results show that the BAM model is relevant for phage-bacterium interactions in *P. aeruginosa* phages, opening possibilities in the clinical settings for choosing phages adapted to mucosal interactions. However, not all phages are positively influenced by mucins (25); therefore, describing the effect of mucosal environment both on phages (via mucin binding) and on the bacterial host physiology (phage susceptibility) is central for phage-bacterium interactions. For example, understanding how different phages infect bacteria in the mucosal environment can help design phage cocktails that target antibiotic-resistant pathogens both within and outside of the mucosa. In the case of *P. aeruginosa*, phages that evolved to infect the host in its virulent form could be associated with mucosal surfaces and the BAM model dynamics, while phages specialized to infect the host in its environmental physiological state would not be affected by mucin. This is an interesting division and shows that different phage types based on mucin response could be selected for specific purposes.

Taken together our data ties *P. aeruginosa* phages to the BAM model and opens possibilities to explore mucosal interactions as a mean to improve phage therapy against this pathogen. From a clinical point of view, the active search and employment of phages adapted to mucosal interactions could likely improve the outcome of treatments, considering that these phages are efficient in killing bacteria in its mucin-induced invasive forms. Besides, phage mucin binding could become a mean to rationally explore prophylactic phage therapy approaches. From an ecological point of view, studying *P. aeruginosa* and its phages behavior in different settings such as in the environment and in mucosal conditions could help better understand opportunistic infections and pathogenicity.

## MATERIALS AND METHODS

### Experimental model and subject details

#### Host (P. aeruginosa) details

The host used for making the phage stocks and the experiments was *P. aeruginosa* strain 573 (29). This bacterium was isolated from bone marrow interstitial fluid from a patient in Georgia in the 1970s. Another host used was *P. aeruginosa* strain PA14 (DSM 19882) for the CRISPR spacer acquisition experiments, for a growth experiment using phage GEC_PNG14 and for testing the mucin effect on biofilm formation. Both hosts were grown using a culture media consisting of peptone (5 g/L) and meat extract (3 g/L). Whenever needed cultures were prepared in a final volume of 5 mL and incubated at 37°C under agitation of 200 rpm in a shaker. The mucin used in this study for the simulation of mucosal conditions was purified porcine mucin (Sigma, catalog no. M1778). Before use, a 2% (wt/vol) stock solution was prepared in water and autoclaved. The stock solution was diluted to the desired concentration in growth media whenever cultures in the presence of mucin were used. To account for the media concentration, control

cultures (no mucin) in these experiments received the same volume of autoclaved water instead of the mucin stock solution.

### Phage details

Four different phages isolated at the Eliava Institute in Tbilisi (Georgia) were used in this study: GEC_PNG3, GEC_PNG14, GEC_MRC, and GEC_K2. Details about these phages are shown in the supplementary material.

Preparation of double agar plates for phage work was made by mixing 0.3 mL of an overnight culture of the bacterial host strain to 3 mL of growth media containing 0.7% (wt/vol) agar (soft-agar). The mixture was briefly vortexed and added to the top of a 1.5% (wt/vol) agar plate. Phages were propagated in the host strain *P. aeruginosa* strain 573. For preparing stocks, a fully confluent double agar plate was prepared, and the top agar layer was collected, mixed with 4 mL of growth media, centrifuged (11,000 *g*, 10 min, Sorvall RC34), and filtered using a 0.22-µm syringe filter.

Phage titrations were made by preparing double agar plates as described above. Phage-containing solutions were serially diluted in growth media, and 10 µL drops of the dilutions was added to the top of freshly made double agar plates containing the appropriated host. Plaque-forming unities (pfu) were counted after incubating the plates for a day at 37°C.

## Method details

### Phage host range determination

All of the four phages used in this study had their host range evaluated against 40 clinical strains of *P. aeruginosa* isolated from patients in Georgia. This was made to test their potential for use in clinics. Host range was determined by spot test analysis using three different concentrations of each phage to account for the effect of the multiplicity of infection (MOI = 1, MOI = 0.1, and MOI = 0.01) on the final results. The appearance of phage plaques on bacterial lawns of the tested clinical strains was considered a sign that the phage infects the host in question.

### Genome sequencing and analysis

The *P. aeruginosa* strain 573 genome has been sequenced previously, and the data are available in Pseudomonas Genome DB under name *Pseudomonas aeruginosa* CN573 = PSE143 (29). We used CRISPRCasFinder (30) to identify CRISPR-Cas systems in the bacterial strain and to extract spacers. Possible prophages were searched with Prophage Hunter (31) and Phaster (32).

For phage genomic sequencing, infections were made using the *P. aeruginosa* strain 573 as host. Confluent soft-agar lawns were harvested, mixed with 4 mL of culture media, centrifuged (11,000 *g*, 10 min, Sorvall RC34), and filtered. Phage precipitation was made with $ZnCl_2$ followed by the removal of host DNA with nucleases (33), with the following adaptation: after protease K treatment, Guanidine:Ethanol (1 part 6 M guanidine and 2 parts 99% ethanol, vol/vol) was added to each sample and the DNA extraction proceeded with the GeneJet Genomic DNA Purification Kit (Thermo Fisher). Paired-end sequencing (150PE) was made using the DNBSEQ platform at BGI.

Phage genomes were *de novo* assembled using Velvet in Geneious Prime (except for GEC_PNG14, which was assembled with Megahit (34), and genome ends and packaging strategy were checked with PhageTerm (35). Open reading frames were predicted with Glimmer (36). BLASTp (37) was employed to annotate ORFs, and carbohydrate- and Ig-binding domains were identified with HMM-HMM comparisons (HHpred with UniRef 100 database) (38). Phyre was used to identify Anti-CRISPR proteins (39). Finally, tRNAs were predicted with Aragorn (40). Clinker (41) was used for genomic comparisons. Average nucleotide identities were calculated using OrthoANIu algorithm (42).

## Phage adhesion to mucin

Adhesion of phages to mucin is the first phenotype associated with the BAM model tested with the phages used in this study. This was made *in vitro* as described previously (18, 20). In these studies, phages shown to interact with mucins had a mean ratio of "phages in mucin plates/phages in control plates" between 2.1 and 7. To test for mucin binding, phages were diluted in culture media to a final concentration of $2.5 \times 10^3$ pfu/mL and 5 mL of the dilutions was added to agar plates supplemented with 1% mucin or without mucin in its composition (control). The plates were kept shaking at room temperature for 30 min, then all the liquid was removed from the plates, and 3 mL of soft-agar containing 0.3 mL of an overnight *P. aeruginosa* strain 573 was added to each plate. Phage plaques were counted after 2 days of incubation at 37°C.

## Phage growth in mucin

The second phenotype associated with the BAM model tested was the effect of mucin for phage growth. We have already seen that pre-exposure of the host to mucin can lead to improved phage growth in some phage-host pairs (20, 22). The effect of mucin to phage growth was tested by adding the *P. aeruginosa* 573 host to cultures containing 0.5× media alone or 0.5× growth media supplemented with 0.1% and 1% of purified porcine mucin. To measure the precise number of cells added, we plated the inoculum and counted colony-forming units (cfu), determining that approximately $6.5 \times 10^3$ cfu were used at time zero. After 3 h of incubating the host in control or mucin conditions, $10^4$ pfu of phages were added. Sampling was made at 2, 22, and 44 h after infection by removing 100 µL of the cultures. Ten microliters of chloroform was added to each sample, and it was followed by serial dilution and phage titration. All incubations were made at 37°C in an orbital shaker (200 rpm).

## Biofilm assay

Although no mechanism behind the improved phage growth in mucin-exposed hosts has been described, it likely happens because of mucin-elicited changes in bacterial physiology. To confirm that our hosts are, indeed, being changed by mucin exposure, we compared the formation of biofilm in both hosts in the presence or absence of mucin. To do so, *P. aeruginosa* strains PA14 and 573 were cultured overnight in LB at 37°C under constant agitation. To analyze their biofilm formation under mucin supplementation, the cultures were inoculated 1:10,000 in LB with 0%, 0.05%, and 0.1% mucin. One hundred fifty microliters of these cultures (and their negative controls without bacteria) was pipetted onto a sterile 96-well plate in six replicates and incubated 48 h at 37°C without shaking. The liquid medium was then discarded, wells washed three times with sterile water, and stained with 0.1% crystal violet (45 min). Excess stains were removed by washing three times with water, and ethanol was added to the wells to solubilize crystal violet. $OD_{595}$ of 100 µL samples was measured with a plate reader (Multiskan FC, Thermo Scientific, China).

## Spacer acquisition experiments

The third phenotype associated with the BAM model tested was whether mucin exposure enhanced CRISPR activity in the host, a phenomenon seen in other phage-bacteria pair in which mucosal conditions affects their interactions (21). Four different experiments were made to test mucin effect on *P. aeruginosa* CRISPR spacer acquisition. For all of these tests, the phage GEC_PNG14 and the *P. aeruginosa* PA14 host were used. The reason for this choice was that GEC_PNG14 grows best in mucin-exposed hosts and can infect *P. aeruginosa* PA14, an already established model to study CRISPR activity in *P. aeruginosa*. The first test consisted of co-culturing the phage and host in culture media without mucin or supplemented with 0.1% mucin at 37°C. The second test consisted of co-culturing the phage and host in autoclaved lake water without mucin or supplemented with 0.1% mucin at 37°C. The third test consisted of co-culturing the phage and host

in 0.1× culture media without mucin or supplemented with 0.1% mucin at 25℃. And, the fourth test consisted of co-culturing the phage and host in autoclaved lake water without mucin or supplemented with 0.1% mucin at 25℃, but with the presence of a competitor bacterium (*Flavobacterium columnare*). All tests were incubated for 1 month under agitation at the indicated temperatures, in 5 mL of the final volume, with weekly samplings.

Sampling consisted of removal and renewal of 1 mL of the cultures. The collected material was serially diluted and plated, and colonies obtained were used as template for PCRs designed to detect spacer acquisition. Primers used to check spacer acquisitions were CR2SPF: GAGGGTTTCTGGCGGGAA plus CR2SPR: GTCCAGAAGTCACCACCCG (43) and CRISPR1F: TTGGGGCTTGGAAGGTTGAT plus CRISPR1R: AAGGCCAGCGCGCCGGTGAT (44). DreamTaq polymerase (Thermo Fisher) was used to prepare the reactions, which contained: 0.5 mM of DNTPs, 0.5 µM of each primer, DreamTaq buffer at 1× and 5 µL of template in a final volume 20 µL. Template preparation consisted of mixing one colony into 50 µL of growth media (direct colony PCR without prior DNA extraction). Cycling conditions were 95 degrees for 3 min followed by 30 cycles of 95 degrees for 30 s, 60 degrees for 30 s, 72 degrees for 1 min, and a final extension step of 72 degrees for 15 min. PCRs were resolved in 2% agarose gels, and the addition of spacers was verified by the size of each amplicon.

## Quantification and statistical analysis

Graphs and statistical analysis were made using Graphpad Prism version 9.3.1 (GraphPad Software, San Diego, California USA, www.graphpad.com). Datapoints are represented either in bar graphs showing each replicate as an individual point or in line graphs where the average value is plotted for each time point tested. The standard deviation of all datapoints is shown in every graph. Conditions were tested in triplicates, and unpaired *t* tests were used to compare controls and tested conditions whenever necessary. A *P* value below 0.05 was considered to be significant, and the precise *P* values are indicated in the text. On some occasions, we compared control to mucin conditions by calculating a ratio. This mucin:control ratios were calculated using the averages from data points. The figure legends contain the information written above for each experiment.

## ACKNOWLEDGMENTS

We would like to thank MSc Noora Rantanen for skillful help. This study was funded by Academy of Finland (#314939 and #346992 for L.-R.S.) and the Centre for New Antibacterial Strategies (CANS) of the Arctic University of Norway (project ID #2520855 for G.M.D.F.A.). This project has received funding from the European Union's Horizon 2020 research and innovation programme under grant agreement No. 767015. The content of this article reflects only the views of its authors. The European Commission is not responsible for any use that may be made of the information it contains.

G.M.D.F.A. and L.-R.S. designed and performed the experiments. N.G., N.B., E.J., E.K., S.M., and N.C. isolated and characterized the phages used. L.-R.S. and J.R. did the bioinformatics analysis. L.-R.S, N.C., and N.P. were responsible for project management and funding acquisition. All authors contributed to the preparation of the manuscript.

## AUTHOR AFFILIATIONS

[1]Faculty of Biosciences, Fisheries and Economics, The Norwegian College of Fishery Science, UiT—The Arctic University of Norway, Tromsø, Norway
[2]Molecular and Integrative Biosciences Research Programme, University of Helsinki, Helsinki, Finland
[3]George Eliava Institute of Bacteriophages, Microbiology & Virology, Tbilisi, Georgia
[4]Faculty of Natural Science and Medicine, Ilia State University, Tbilisi, Georgia
[5]University of Manchester, Manchester, United Kingdom

⁶Department of Biological and Environmental Science and Nanoscience Centre, University of Jyväskylä, Jyväskylä, Finland

## AUTHOR ORCIDs

Gabriel Magno de Freitas Almeida ⓘ http://orcid.org/0000-0003-2317-5092
Lotta-Riina Sundberg ⓘ http://orcid.org/0000-0003-3510-4398

## FUNDING

| Funder | Grant(s) | Author(s) |
| --- | --- | --- |
| Research Council of Finland (AKA) | 346992 | Lotta-Riina Sundberg |
| EC \| Horizon 2020 Framework Programme (H2020) | 767015 | Nina Chanishvili |
| | | Nikolaos Papadopoulos |

## AUTHOR CONTRIBUTIONS

Gabriel Magno de Freitas Almeida, Conceptualization, Data curation, Formal analysis, Investigation, Methodology, Project administration, Validation, Visualization, Writing – original draft, Writing – review and editing | Janne Ravantti, Conceptualization, Data curation, Formal analysis, Funding acquisition, Investigation, Methodology, Project administration, Resources, Supervision, Validation, Visualization, Writing – original draft, Writing – review and editing | Nino Grdzelishvili, Data curation, Formal analysis, Investigation, Methodology, Resources, Validation, Visualization, Writing – original draft, Writing – review and editing | Elene Kakabadze, Data curation, Formal analysis, Funding acquisition, Investigation, Methodology, Resources, Writing – original draft, Writing – review and editing | Nata Bakuradze, Data curation, Formal analysis, Investigation, Methodology, Resources, Writing – original draft, Writing – review and editing | Elene Javakhishvili, Data curation, Formal analysis, Investigation, Methodology, Resources | Spyridon Megremis, Data curation, Formal analysis, Investigation, Methodology, Resources | Nina Chanishvili, Data curation, Formal analysis, Funding acquisition, Investigation, Methodology, Resources, Writing – original draft, Writing – review and editing | Nikolaos Papadopoulos, Data curation, Formal analysis, Investigation, Methodology, Resources | Lotta-Riina Sundberg, Conceptualization, Data curation, Formal analysis, Funding acquisition, Investigation, Methodology, Project administration, Resources, Supervision, Validation, Visualization, Writing – original draft, Writing – review and editing

## DATA AVAILABILITY

The genomes of the four phages used in this study are available in GenBank under the accession numbers PP836134 (GEC_K2), PP836135 (GEC_MRC), PP836136 (GEC_PNG3), and PP836137 (GEC_PNG14).

## ADDITIONAL FILES

The following material is available online.

### Supplemental Material

**Figure S1 (Spectrum03520-23-s0001.tif).** Imaging and host range of the phages used in this study.
**Figure S2 (Spectrum03520-23-s0002.tif).** Genomic organization and comparison of *Pseudomonas aeruginosa* phages.
**Supplemental text (Spectrum03520-23-s0003.docx).** Characterization of the phages, host used in this study, and Tables S1 and S2.

**Table S1 (Spectrum03520-23-s0004.xlsx).** Annotations of *Pseudomonas aeruginosa* phage GEC_MRC.

**Table S2 (Spectrum03520-23-s0005.xlsx).** Annotations of *Pseudomonas aeruginosa* phage GEC_PNG3.

**Table S3 (Spectrum03520-23-s0006.xlsx).** tRNA predictions based on ARAGORN.

**Table S4 (Spectrum03520-23-s0007.xlsx).** Annotations of *Pseudomonas aeruginosa* phage GEC_K2.

**Table S5 (Spectrum03520-23-s0008.xlsx).** Annotations of *Pseudomonas aeruginosa* phage GEC_PNG14.

**Table S6 (Spectrum03520-23-s0009.xlsx).** CRISPR Spacer hits.

**Table S7 (Spectrum03520-23-s0010.xlsx).** Prophage annotation by PHASTER.

## Open Peer Review

**PEER REVIEW HISTORY (review-history.pdf).** An accounting of the reviewer comments and feedback.

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
