## [Reviewer comments · Microbiology Spectrum]

Microbiology Spectrum

Relevance of the bacteriophage adherence to mucus model for *Pseudomonas aeruginosa* phages

Gabriel Almeida, Janne Ravantti, Nino Grdzelishvili, Elene Kakabadze, Nata Bakuradze, Elene Javakhishvili, Spyridon Megremis, Nina Chanishvili, Nikolaos G. Papadopoulos, and Lotta-Riina Sundberg

Corresponding Author(s): Gabriel Almeida, UiT Norges arktiske universitet

Review Timeline:

Submission Date:	September 29, 2023
Editorial Decision:	December 9, 2023
Revision Received:	March 5, 2024
Accepted:	May 13, 2024

Editor: Daria Van Tyne

Reviewer(s): Disclosure of reviewer identity is with reference to reviewer comments included in decision letter(s). The following individuals involved in review of your submission have agreed to reveal their identity: Jeremy J Barr (Reviewer #2)

Transaction Report:

DOI: <https://doi.org/10.1128/spectrum.03520-23>

Re: Spectrum03520-23 (Relevance of the bacteriophage adherence to mucus model for Pseudomonas aeruginosa phages)

Dear Dr. Gabriel MF Almeida:

Thank you for the privilege of reviewing your work. Your manuscript was reviewed by two experts and I would now like you to revise your study in line with their feedback. Below you will find instructions from the Spectrum editorial office and the reviewer comments.

Revision Guidelines

Sincerely,
Daria Van Tyne
Editor
Microbiology Spectrum

Reviewer #1 (Comments for the Author):

Review of Relevance of the bacteriophage adherence to mucus model for Pseudomonas aeruginosa phages
Summary:

Authors examine the effects of mucin on 4 bacteriophages. The result shows the mucin had some increased effect on the reported PFU of phages. This was tested in two ways. A modified EOP assay in which phage were allowed to bind to a mucin containing agar plate. A mucin exposure assay in which bacteria were pre exposed to mucin. The paper also describes the 4

novel phages as well as explores if resistance via CRISPR-cas is promoted in the presence of mucin for a single phage.

In keeping with the ethos of Spectrum I believe this work is scientifically rigorous. I would argue that the explanation and experimentation require consideration. Most notably, the assays to determine phages affinity for mucin could be the result of other factors. For instance, the addition of mucin could stabilize the phage or change their orientation which could induce a higher efficiency of plating. The authors should consider a non-host associated method of quantifying the phage.

In a similar vein, the mucin induction assay has other alternative hypotheses not explored in the paper. I find the inclusion of mucin increasing growth rate confusing as it is known that pseudomonas does not use mucin for growth in the absence of other bacterial species. Mucin could change the biogeography of bacteria increasing the likelihood of an effective subsequent infection. Essential if the hosts of spatially structured in such a manner to improve phage predation we might expect the same results.

A further alternative is the measurement increase is due to an increase in the free phage and not a measurement of phage replication. Examining the infection dynamics of phage, i.e. how many bacteria were killed by phage would provide some insight into if the phage are more infective under mucin or more likely to be counted.

While these two points are my subjective opinions on the matter there are a couple of points I think should be addressed prior to publication.

First why did GEC_PNG3 grow so well in under 2 hours? See figure 4c. Can the authors check the burst size and eclipse period of this phage. This problem is compounded by the work shown in figure 5C. under 0 mucin concentration the PFU reported appears lower than the added PFU.

Data presented from line 223-231 requires statistical analysis.

As an editorial note, while I appreciate the characterization of the phage presented in the paper it does little to advance the hypothesis been tested. I would move most of this to an SI section as it currently impedes the flow of the paper.

Reviewer #2 (Comments for the Author):

The manuscript by Almeida et al., reports a preliminary characterisation of four virulent anti-Pseudomonas phages and characterises these phages in relation to the BAM model. The authors report that three out of these four phage report BAM-associated phenotype. They associated these with increased mucin adherence, increased virulence on mucus-associated hosts, and impact of CRISPR mechanisms. The writing in the paper could be cleaned up some more, there were several transition sentences and statements that were not well written. I have a few comments the authors should address.

Abstract

Line 35 - Use of double passive - just use hypothesize

Line 44 - I have several comments on this claim (see below), but the use of 'likely' anti-CRISPR mechanisms is too strong of a claim with little supporting evidence

Introduction

Lines 74-76 - Should broaden this statement to something like 'or by phage therapy centers, such as the Eliava'

Results

Line 129 - There are no plaque images in Fig1A

Line 156 - It would make the paper flow better if you first introduce the sequences of the phages, before the CRISPR analysis of the host and its matching spacers

Line 175 - Given the focus on the BAM model for these phages, I'm surprised that no attempts to find/match with structural Ig-like domains, which are suggestive of mucus-adherence, were attempted.

The authors should run additional informatic analyses to look for HMM and structural predictions of Ig-like domains, and potentially other carbohydrate-binding folds, in the structural gene components of their four phage

Line 195 - How did you determine they are lytic? Or more appropriately, how did you determine they do not exhibit lysogeny?

Lines 219-221 - see prior comment - no mention of the analyses performed. How detailed was your search for alternative mucin-binding domains?

Line 285 - This is too strong a conclusion. You need evidence to back up this assumption. I would suggest the authors present the data associated with CRISPR repeat expansion assay (were any new repeats found in the CRISPR array?). Was any phage resistance seen in this experiment? The authors should further review some of Edze Westra work showing the preferential selection of surface-based mutations over CRISPR mechanisms in mono-culture assays (I recognise one co-culture was performed here, but this does not absolve the point).

Further, they should either provide evidence that PNG14 contains predicted anti-CRISPR genes, or remove/tone back this claim.

Lines 329-331 - Overstatement on Anti-CRISPRs

Rebuttal letter to the reviewers' comments:

Reviewer #1 (Comments for the Author):

Review of Relevance of the bacteriophage adherence to mucus model for *Pseudomonas aeruginosa* phages

Summary:

Authors examine the effects of mucin on 4 bacteriophages. The result shows the mucin had some increased effect on the reported PFU of phages. This was tested in two ways. A modified EOP assay in which phage were allowed to bind to a mucin containing agar plate. A mucin exposure assay in which bacteria were pre exposed to mucin. The paper also describes the 4 novel phages as well as explores if resistance via CRISPR-cas is promoted in the presence of mucin for a single phage.

Answer 1: We thank you for your time and comments. We noticed that you divided your comments into the first classified by you as “subjective opinions” and the rest as points that should be addressed for the publication. Your subjective points are very important, and we did our best to provide background here in this letter to discuss them with you. These are our answers 2-4 below. Then we proceeded with answering your other points.

In keeping with the ethos of Spectrum I believe this work is scientifically rigorous. I would argue that the explanation and experimentation require consideration. Most notably, the assays to determine phages affinity for mucin could be the result of other factors. For instance, the addition of mucin could stabilize the phage or change their orientation which could induce a higher efficiency of plating. The authors should consider a non-host associated method of quantifying the phage.

Answer 2: Thank you for considering our work scientifically rigorous. We agree that the studies focused on phage-mucin interactions would benefit greatly from methods aimed at observing or measuring the interactions between phage particles and mucin fibers that are independent of titration experiments. Since 2016 we are working with phages from a mucosal perspective and attempting to find alternative ways to measure/quantify phage-mucin interactions. Here are some highlights of our attempts, most using *Flavobacterium* phages known to have affinity to mucin:

- Direct imaging has been impossible given the frail structure of mucosal layers. However, when phages were added to flowthrough aquaria they have been consistently re-isolated from fish skin mucus and sporadically from the water for several days without the host in the system.
- We have made tests concerning the stability of phage solutions with or without mucin added. No differences in titres were measured in stocks stored in cold for several weeks. When a mucin-binding phage was incubated with mucin (under agitation, 25°C), as in the case of our phage-only controls from Almeida et al 2022 (<https://doi.org/10.1038/s41467-022-31330-3>), the phage was gone after a week in the phage-only tubes but persisted for months in the presence of the host, with or without mucin added.
- We also tried to develop pull-down assays using mucin and phages, either by centrifugating mucin-phage solutions or letting them slowly precipitate for many hours or even days. All collected fractions had roughly the same phage counts in every experiment. The reason being that likely the interactions between phage Ig-like domains and mucin fibers are transient and weak.

Because of our inability to find a better method to rank our phages in relation to mucin affinity, we kept using the traditional plate methodology to evaluate our *Pseudomonas* phages affinity to mucin. The methodology we used is an adaptation of the protocol used in the original BAM model paper (Barr et al 2013, <https://doi.org/10.1073/pnas.1305923110>). The main difference is that we add mucin already to the agar preparation instead of letting a mucin-containing solution dry on top of the plate prior to phage addition. This has been extensively done to test our phage collections, as seen in our previous publications (Almeida et al 2019, <https://doi.org/10.1128/mbio.01984-19>). We agree that it is not a perfect model, especially considering the high phage background in control (no-mucin) plates, but it is enough for conveniently test phages and compare the affinity phenotype to others tested with the same method. We had written about this protocol and provided some values for comparison of the ratio between 'phages in mucin plates / phages in control plates' found in this manuscript methods.

In a similar vein, the mucin induction assay has other alternative hypotheses not explored in the paper. I find the inclusion of mucin increasing growth rate confusing as it is known that *Pseudomonas* does not use mucin for growth in the absence of other bacterial species. Mucin could change the biogeography of bacteria increasing the likelihood of an effective subsequent infection. Essential if the hosts of spatially structured in such a manner to improve phage predation we might expect the same results.

Answer 3: Like the issue of measuring phage affinity to mucin, refined methods to understand why mucin presence affects phage growth would also be welcomed. If not mistaken, the first direct observation that mucin presence affected phage growth was made by us in 2019 (Almeida et al 2019, <https://doi.org/10.1128/mbio.01984-19>). In that paper we have shown that the phenotype was true for *Flavobacterium* and *Aeromonas* phages. At the time one other paper had shown that a *Clostridium difficile* phage activity was improved in the presence of eukaryotic cells (Shan et al 2018, 10.1038/s41598-018-23418-y). Although the mucin hypothesis was not considered by the authors, the eukaryotic cells used are known to constitutively express at least one mucin gene. In 2019 an *Escherichia coli* phage 'enhanced' by mucin presence was described (Green et al 2021, <https://doi.org/10.1128/mbio.03474-20>). Then in 2022 we detected that a *Streptococcus mutans* phage also benefited from mucin presence, since phage replication was only detected in mucin-containing cultures (Sundberg et al 2022, <http://doi.org/10.1089/phage.2022.0021>). The papers mentioned in this paragraph are cited in our manuscript.

One explanation for improved phage growth in mucin conditions could be that mucin serves as nutrient and thus increases the host population. However, this does not seem to be the case. As you mentioned, *Pseudomonas aeruginosa* does not use mucin as nutrient source. We have stated this in our introduction and shown this experimentally with our test strain in Figure 5A (now figure 3A in the revised manuscript). Mucin affecting biogeography in the samples is another explanation, even used by Green et al 2021 to explain their *E. coli* phage enhanced by mucin. However, as discussed in the point above, observing phage orientation and bacteria-mucin dynamics is a tricky process. Besides, all our infection experiments are made with cultures under agitation, which would likely affect weaker structural interactions like those predicted for phage-mucin.

Our other models have given us a better indication of the mechanisms behind improved phage growth under mucosal conditions. It is clear that mucin, despite not being a nutrient source, may affect bacterial physiology. In the case of *Flavobacterium* and *Aeromonas* a switch to biofilm formation is clear. We went deeper into the *Flavobacterium* model and described that mucin

exposure upregulates several virulence factors, increases *in vivo* virulence, and even favour CRISPR activity (Almeida et al 2022 ; <https://doi.org/10.1038/s41467-022-31330-3>). Our hypothesis is that mucin leads to physiological changes in pathogenic bacterial species that invades mucosal layers. From the bacterial side, these changes are used to adopt an invasive phenotype. From the phage side, evolution has favoured infection mechanisms that exploits the mucin-enhanced bacterial hosts. This adds one more co-evolution layer to the BAM model. We discuss physiological changes after mucin exposure leading to establishment of invasive phenotypes in *P. aeruginosa* in our introduction. We discussed our hypothesis of phages exploiting this for affecting phage growth in our discussion, citing the literature supporting the claim.

A further alternative is the measurement increase is due to an increase in the free phage and not a measurement of phage replication. Examining the infection dynamics of phage, i.e. how many bacteria were killed by phage would provide some insight into if the phage are more infective under mucin or more likely to be counted.

Answer 4: We did not measure the stability of our *Pseudomonas* phages in free phage conditions. But as mentioned above, when this was made for *Flavobacterium* phages, the phage inoculum disappeared in a few days in the absence of hosts but remained detectable for many weeks in co-cultures with the host. In Figure 4A-D (now figure 2A-D in the revised manuscript), we measured phage growth of all four phages and it is clear to see that the titres are higher than the inoculum in the time points tested, showing that we are indeed measuring the phage progeny and not only re-isolating the inoculum.

In Figure 5B (now figure 3B in the revised manuscript), we tested the hypothesis of whether different phage-host ratios (multiplicity of infection) would explain the differences in phage replication using phage PNG3 and the Eliava *P. aeruginosa* host. What we saw is that, regardless of a low or high multiplicity of infection, the phage produces more progeny in mucin-containing cultures. Then, in Figure 5C (now figure 3C in the revised manuscript), we measured the *P. aeruginosa* PA14 host cfu counts and the phage PNG14 pfu counts from the same cultures. What we saw is that bacteria numbers do not differ between mucin and no-mucin conditions, while phage titres are considerably higher in the presence of mucin.

We must also consider that before every phage titration all samples were treated with 10% chloroform. This means that we are measuring the whole phage population in every condition tested, and not excluding phages 'trapped' in the host cells or favouring only free phages from the cultures.

While these two points are my subjective opinions on the matter there are a couple of points I think should be addressed prior to publication. First why did GEC_PNG3 grow so well in under 2 hours? See figure 4c. Can the authors check the burst size and eclipse period of this phage. This problem is compounded by the work shown in figure 5C. under 0 mucin concentration the PFU reported appears lower than the added PFU.

Answer 5: Thank you for this observation. Indeed, phage PNG3 differs from the others in the first time point shown in Figure 4C (now figure 2C in the revised manuscript). In this experiment titres were high in all conditions tested after 2 hours of infection. It is not an issue with our phage inoculum or miscalculation of the stock titre: we got the expected number of plaques in the phage adhesion experiment using the same stock. Also, for the experiment in which this phage was used as model to

test the multiplicity of infection effect (Figure 5B, now figure 3B in the revised manuscript), we have quantified the phage inoculum used for both the high and the low moi, confirming the stock titre. It is also important to note that phage PNG3 produces the large plaques and was one of the most effective against a varied amount of *P. aeruginosa* clinical isolates. When tested in an independent experiment, PNG3 latent period was approximately 27 minutes, showing that indeed it is a fast-growing virus.

Regarding your comment on Figure 5C (now figure 3C in the revised manuscript), the amount of PNG14 added as 1×10^5 PFU, which is higher than the titre measured in the 0% mucin concentration. But we believe that you are referring to Figure 5B (now figure 3B) in the revised manuscript, right? The added amount of PNG3 to this experiment was 9.9×10^5 PFU (high MOI) and 9.9×10^1 PFU (low MOI) to the no-mucin conditions. The samples were quantified 44h after infection, and the titres are close to the inoculum likely because there is a quicker decrease in titre for PNG3 in the no-mucin conditions (as seen in the growth curves from Figure 4C, now figure 2C in the revised manuscript).

Data presented from line 223-231 requires statistical analysis.

Answer 6: Sorry for not indicating the p values to support this affirmation. We have provided it in the text now. It now reads: *“All the four phages tested grew in their *P. aeruginosa* 573 host, as expected (Figure 2A-D). However, two of these phages grew more efficiently when the host was exposed to mucin for three hours before infection. Phage GEC_PNG3 titre grows then decreases in control conditions, but increases and remains high at the 44 hour time point if the host was exposed to mucin ($p=0.0164$) (Figure 2C and 2G). Phage GEC_PNG14 also grew best in mucin-exposed hosts (Figure 2D and 2H), but with a growth peak at the 22 hour time point ($p=0.0014$)”*.

As an editorial note, while I appreciate the characterization of the phage presented in the paper it does little to advance the hypothesis been tested. I would move most of this to an SI section as it currently impedes the flow of the paper.

Answer 7: Thank you for this comment. After considering the paper structure we agree that the phage collection characterization, despite being important, does little to improve the BAM model dynamics tested which are the highlight of the publication. Because of this and agreeing with you we have moved the phage characterization material to the supplementary section, leaving a modified version of the *“Details about the phages used in this study”* topic in the results referencing the supplementary information. Note that this change altered the figure numbering on the main manuscript.

Reviewer #2 (Comments for the Author):

The manuscript by Almeida et al., reports a preliminary characterisation of four virulent anti-*Pseudomonas* phages and characterises these phages in relation to the BAM model. The authors report that three out of these four phage report BAM-associated phenotype. They associated these with increased mucin adherence, increased virulence on mucus-associated hosts, and impact of CRISPR mechanisms. The writing in the paper could be cleaned up some more, there were several transition sentences and statements that were not well written. I have a few comments the authors should address.

Answer 8: We thank you for your time and for your comments. While modifying our manuscript we took care to improve the written language and clean some of the transition sentences. We will answer your comments one by one below.

Abstract

Line 35 - Use of double passive - just use hypothesize

Answer 9: It was corrected.

Line 44 - I have several comments on this claim (see below), but the use of 'likely' anti-CRISPR mechanisms is too strong of a claim with little supporting evidence

Answer 10: Thank you for your comment. We have modified this claim. It now reads "*We could not detect CRISPR activity in our system but identified two putative anti-CRISPR proteins coded by the phage*".

Introduction

Lines 74-76 - Should broaden this statement to something like 'or by phage therapy centers, such as the Eliava'

Answer 11: It was corrected.

Results

Line 129 - There are no plaque images in Fig1A

Answer 12: Thank you for noticing this. We had the plaque images in our draft figure 1 but decided to remove them before submission, since all plaques were similar despite small size differences and the images were not bringing relevant information. We corrected the text. Please note that we decided to follow reviewer #1 suggestion to move the phage characterization part of the manuscript to the supplementary material, and this changed the figure numbering order.

Line 156 - It would make the paper flow better if you first introduce the sequences of the phages, before the CRISPR analysis of the host and its matching spacers

Answer 13: Thank you for this comment, we have reordered the phage characterization section to improve its flow. It is also important to note that, in agreement with reviewer #1, we moved this section to the supplementary material for making the main text with a better flow focused on the BAM model dynamics.

Line 175 - Given the focus on the BAM model for these phages, I'm surprised that no attempts to find/match with structural Ig-like domains, which are suggestive of mucus-adherence, were

attempted.

The authors should run additional informatic analyses to look for HMM and structural predictions of Ig-like domains, and potentially other carbohydrate-binding folds, in the structural gene components of their four phage

Answer 14: Thank you for the suggestion. We have now performed additional bioinformatic analyses using HMM-HMM comparisons (HHpred with UniRef 100 database) and structural comparisons using Phyre. We especially focused in finding carbohydrate-binding domains and Ig-like domains. HMM search indeed produced high quality hits (> 98% probability) to Ig-like and carbohydrate binding domains for all phages (see table below), these are now mentioned in the text.

Response letter table 1: Identified Ig-binding and carbohydrate -binding domains (ORF number, locus tag).

Phage	ORF (locus tag) with Ig-like domain hit (probability, e-value)	ORF (locus tag) with carbohydrate binding domain hit (probability, e-value)
PNG14	ORF51 (GEC_PNG14_051): tail protein (99.8%, 4.4E-24)	ORF48 (GEC_PNG14_048) (99.3%, 1E-14)
K2	ORF20 (GEC_K2_019): hypothetical protein (with TIGR02594 domain) (99.89%, 3.8E-27) ORF34 (GEC_K2_032): major tail tube protein (99.96%, 2.5E-33) ORF43 (GEC_K2_039): Central tail hub protein (99.96%, 3.1E-33)	ORF9 (GEC_K2_008): hypothetical protein (99.61%, 1.2E-18)
MCR	ORF100 (GEC_MCR_097): hypothetical protein (96.88%, 9.4E-05) ORF117 (GEC_MCR_114): hypothetical protein (96.02%, 0.0022)	ORF96 (GEC_MCR_094): Putative tail fiber protein (99,5%, 9.5E-13) ORF99 (GEC_MCR_096): Putative tail fiber protein (99,5%, 9.5E-13)
PNG3	ORF101 (GEC_PNG3_096): hypothetical protein (98,18%, 2.6E-08) ORF118 (GEC_PNG3_113): hypothetical protein (96%, 0.0022)	ORF97 (GEC_PNG3_093): putative tail fiber protein (99,05%, 9.5E-13) ORF100 (GEC_PNG3_095): putative tail fibre protein (99,71%, 1.1E-20)

Line 195 - How did you determine they are lytic? Or more appropriately, how did you determine they do not exhibit lysogeny?

Answer 15: Of the four phages, only GEC_PNG14 had genetic markers of lysogeny. However, from our experience combined (experiments using these phages made in Finland, in Georgia and in the UK), no

sign of lysogeny in experimental conditions was detected. This includes the work done during the isolation process, preparation of stocks, host-range analysis and the experiments described here. One highlight besides the consistency of titres, is that plaques were always clear and not turbid. We have made this clear with the sentence “*lytic in the laboratory conditions used in the described experiments*” in the manuscript.

Lines 219-221 - see prior comment - no mention of the analyses performed. How detailed was your search for alternative mucin-binding domains?

Answer 16: As mentioned above, we have performed more analyses and found hits to Ig-like domains and carbohydrate-binding domains. The detailed search criteria is now included in the manuscript.

Line 285 - This is too strong a conclusion. You need evidence to back up this assumption. I would suggest the authors present the data associated with CRISPR repeat expansion assay (were any new repeats found in the CRISPR array?). Was any phage resistance seen in this experiment? The authors should further review some of Edze Westra work showing the preferential selection of surface-based mutations over CRISPR mechanisms in mono-culture assays (I recognise one co-culture was performed here, but this does not absolve the point).

Answer 17: Thank you for the observation. We are aware of Westra’s papers and were inspired by them, mixing some of the conditions tested by his group with our methodology applied to *Flavobacterium columnare* and mucosal conditions from Almeida et al 2022 (<https://doi.org/10.1038/s41467-022-31330-3>). The readouts of all four experiments were: phage titres (from titrations using an aliquot of the cultures), bacteria titres (from platings using an aliquot of the cultures), and spacer acquisition data (from testing random colonies by PCRs). The reason for not showing the data is that no acquisition at all happened despite both phage and bacteria populations remaining viable in the cultures. In other words, no new spacer appeared in any sample during any sampling.

Since the common factor between all the experiments were the phage and host used, and considering that we had detected putative anti-CRISPR orfs in the phage used, we thought that likely our phage is not the most suited for this type of analysis. Testing other phages and other *P. aeruginosa* strains, especially those shown by Westra to be involved in CRISPR responses, is an interesting perspective and likely deserve one publication of its own. To avoid such a strong conclusion in the text, we removed the “*and the likely reason is the presence of putative anti CRISPR mechanisms on phage GEC_PNG14*” part of this sentence, leaving only the statement that no spacer acquisition was detected in our four different experiments.

Further, they should either provide evidence that PNG14 contains predicted anti-CRISPR genes, or remove/tone back this claim.

Answer 18: Bioinformatic analysis suggests two putative Anti-CRISPR proteins in PNG14. ORF31 (locus tag: GEC_PNG14_031) was annotated as AcrIF2 in the initial BlastP analysis, and additional HHPred analysis (using PDB database) gave a hit to Pseudomonas phage D112 anti-CRISPR protein 30 (Probability 100, E-value 1.7e-51). Interestingly, also the following ORF (locus tag: GEC_PNG14_032) had a hit to anti-CRISPR AcrIE4 in blastP, and also a 100% confidence hit to the same acr type in Phyre

(96% Alignment coverage). This information has been added to the manuscript. These putative anti-CRISPR proteins have not been experimentally verified.

Lines 329-331 - Overstatement on Anti-CRISPRs

Answer 19: Thank you for the observation. We modified the text to avoid this overstatement.

Re: Spectrum03520-23R1 (Relevance of the bacteriophage adherence to mucus model for Pseudomonas aeruginosa phages)

Dear Dr. Gabriel MF Almeida:

Your manuscript has been accepted, and I am forwarding it to the ASM production staff for publication. Your paper will first be checked to make sure all elements meet the technical requirements. ASM staff will contact you if anything needs to be revised before copyediting and production can begin. Otherwise, you will be notified when your proofs are ready to be viewed.

Sincerely,
Daria Van Tyne
Editor
Microbiology Spectrum

Reviewer #1 (Comments for the Author):

The authors have addressed my concerns and produced a nice manuscript.

Reviewer #2 (Comments for the Author):

Thank you for the corrections and further explanations. All my concerns and comments have been addressed.